# Copper Accumulation Efficiency in Different Recombinant Microorganism Strains Available for Bioremediation of Heavy Metal-Polluted Waters

**DOI:** 10.3390/ijms24087575

**Published:** 2023-04-20

**Authors:** Constantina Bianca Vulpe, Mariana Adina Matica, Renata Kovačević, Daniela Dascalu, Zoran Stevanovic, Adriana Isvoran, Vasile Ostafe, Gheorghița Menghiu

**Affiliations:** 1Advanced Environmental Research Laboratories, Department of Biology–Chemistry, West University of Timisoara, Oituz 4A, 300086 Timisoara, Romania; constantina.vulpe92@e-uvt.ro (C.B.V.); mariana.matica@e-uvt.ro (M.A.M.); daniela.dascalu@e-uvt.ro (D.D.); adriana.isvoran@e-uvt.ro (A.I.); vasile.ostafe@e-uvt.ro (V.O.); 2Institute for Advanced Environmental Research, Department of Biology–Chemistry, West University of Timisoara, Oituz 4C, 300086 Timisoara, Romania; 3Mining and Metallurgy Institute, Zeleni Bulevar 35, 19210 Bor, Serbia; renata.kovacevic@irmbor.co.rs (R.K.); zoran.stevanovic@irmbor.co.rs (Z.S.)

**Keywords:** *Escherichia coli*, *Saccharomyces cerevisiae*, growth rate, bioaccumulation

## Abstract

The aim of this research was to investigate the bioremediation conditions of copper in synthetic water. In the present study, copper ions accumulation efficiency was determined using various genetically modified strains of *Saccharomyces cerevisiae* (EBY100, INVSc1, BJ5465, and GRF18), *Pichia pastoris* (X-33, KM71H), *Escherichia coli* (XL10 Gold, DH5α, and six types of BL21 (DE3)), and *Escherichia coli* BL21 (DE3) OverExpress expressing two different peroxidases. Viability tests of yeast and bacterial strains showed that bacteria are viable at copper concentrations up to 2.5 mM and yeasts up to 10 mM. Optical emission spectrometry with inductively coupled plasma analysis showed that the tolerance of bacterial strains on media containing 1 mM copper was lower than the tolerance of yeast strains at the same copper concentration. The *E. coli* BL21 RIL strain had the best copper accumulation efficiency (4.79 mg/L of culture normalized at an optical density of 1.00), which was 1250 times more efficient than the control strain. The yeast strain *S. cerevisiae* BJ5465 was the most efficient in copper accumulation out of a total of six yeast strains used, accumulating over 400 times more than the negative control strain. In addition, *E. coli* cells that internally expressed recombinant peroxidase from *Thermobifida fusca* were able to accumulate 400-fold more copper than cells that produced periplasmic recombinant peroxidases.

## 1. Introduction

The availability of drinkable water is an essential feature for the prevention of diseases and the improvement of the quality of human and animal life. The metabolism of the body and enzymatic regulation in the cells depend on water. Natural waters contain various types of impurities and pollutants (dyes, metals, pharmaceuticals, fluoride, pesticides, and pathogens) derived from agricultural activities, mining activities, soil leaching, rock degradation, rainwater, dissolved aerosol particles from the air, and other human activities, including the processing and use of metal-based materials [1]. The mining industry is one of the main processes leading to the economic development of a country. However, mining generates water and soil pollution in the areas around the mines. Monitoring water and soil and providing remediation methods followed by metal recovery, without disturbing mining activities, are some of the ideal solutions to support a balance between economic, environmental, and human health requirements. Copper is a common element present in natural waters, resulting from mining activities as well as from corrosion of copper pipes or fittings [2]. In copper-rich waters, there is often a high concentration of iron and other metals. For instance, Romania and Serbia are countries with a copper mining industry, where mining areas and waters are under the control of researchers. Copper levels in river waters in mining areas range from 0.0136–0.1158 mg/L in Moldova Noua, Romania, to <0.005–318.7 mg/L in Bor, Serbia [3,4].

Copper is an important and essential mineral for human health as well as for animal and plant growth. It is necessary for normal metabolic processes and when bound to certain enzymes, such as ceruloplasmin, superoxide dismutase, cytochrome c-oxidase, lysyl oxidase, and monoamine oxidase, to act as a catalyst to support many body functions [4,5]. Although it is an essential micronutrient and is needed in the body in very small amounts, excess copper in the body can cause nausea, vomiting, diarrhea, stomach cramps, cirrhosis of the liver with periods of hemolysis, and damage to the renal tubules, brain, and other organs. Chronic copper intoxication leading to liver failure has been reported at 30–60 mg/day for three years, with the lethal dose of copper being about 10–20 g. Persons with Wilson’s disease, a rare genetic condition, are more sensitive to the effects of copper [6].

Remediation of heavy metals in water using various adsorbents is one of the main methods investigated in research laboratories. The most explored adsorbent materials are vegetable wastes (wood, wheat bran, wheat straw, seed husks, fruit and vegetable peels, sugar beet pectin gels, and carrot residues) [7,8,9], animal wastes (fish bones, crab shells, cow, donkey, chicken, horse skeletons, and human hair) [10,11], activated carbon (lignite, coal, biochar, peat, and activated carbon residue from biomass gasification) [11,12], nanomaterials (carbon nanotubes, graphene oxide, nanometer-sized TiO_2_, and nanocomposites) [13,14], natural materials (fly ash, zeolites, clay minerals, and diatomite) [15,16], biopolymers (cellulose, starch, alginate, and chitosan) [17,18], coordination compounds (halometalates and 3D metal-organic frameworks) [19,20,21], fungal and bacterial nonviable biomass [22,23]. The adsorption capacity of heavy metals depends on several factors, such as pH, adsorbent dosage, initial metal concentration, contact time, stirring speed, competing ions, and temperature [24]. Metal accumulation by microbial strains, or bioremediation, has received significant attention in recent years due to the potential use of microorganisms to clean metal-polluted water. Bioremediation is similar to the remediation of heavy metals in water using adsorbents but uses various viable organisms, such as bacteria, fungi, algae, and plants as key tools in treating heavy metals in the environment. It is a non-toxic, cost effective, low technology, and environmentally friendly technique for cleaning up areas contaminated with a wide range of pollutants. The purpose of bioremediation is to stimulate microorganisms with nutrients and other chemicals that allow them to remove pollution. Microorganisms enzymatically attack pollutants and convert them into non-toxic products. They may be autochthonous to the affected area or they may be isolated from other areas and introduced to the contaminated sites [25]. Moldova Noua mining area is under our research, where in river waters were found mainly nonspecific coliforms, diatoms, algae, and protozoa species. Bacteria, yeast, and algae can remove heavy metals from aqueous solutions in substantial quantities. To survive under metal-stressed conditions, microorganisms have evolved mechanisms for the removal of heavy metals that include the efflux of metal ions outside the cell, the accumulation, and formation of a complex with metal ions inside the cell and later reducing the toxic metal ions to a non-toxic state. The microorganisms involved in this process belong to the genera *Bacillus*, *Pseudomonas*, *Streptomyces*, *Rhizopus*, *Aspergillus*, and *Saccharomyces*. Yeast strains and genetically modified bacterial strains that are designed to produce recombinant metalloproteins are generally recognized as safe for humans (GRAS) and are, therefore, even more recommended for the remediation of heavy metal-contaminated water [26,27].

In order to develop an operational, cost-effective, and environmentally friendly technique for cleaning copper-polluted water that could be applied in polluted mining areas in Serbia and Romania (but not only), bioaccumulation experiments were carried out using different recombinant yeast strains (*Saccharomyces cerevisiae* EBY100, INVSc1, BJ5465, GRF18, and *Pichia pastoris* X-33, KM71H) and bacteria (*Escherichia coli* strains XL10 Gold, DH5α, and BL21 (DE3), BL21 (DE3) OverExpress, BL21 (DE3) RP, BL21 (DE3) RIL, BL21 (DE3) Star, and BL21 (DE3) Rosseta Gami, BL21 (DE3) Shuffle) on copper synthetic solutions. The copper accumulation efficiency of an *Escherichia coli* BL21 (DE3) OverExpress expressing two different recombinant peroxidases was also determined. The novelty of this study is based on the development of an effective method of bioremediation of copper-polluted water using genetically modified strains of bacteria and yeast, tested for the first time for their bioaccumulation characteristics. Furthermore, the study showed the potential of using strains that produce two recombinant peroxidases, metal-dependent enzymes, for bioremediation.

## 2. Results and Discussions

### 2.1. Tolerance of Yeast and Bacterial Strains on Copper-Containing Media

The copper tolerance of four *Saccharomyces cerevisiae* strains, two *Pichia pastoris* strains, and nine *Escherichia coli* strains was first assessed by a drop test on Yeast-Peptone-Dextrose (YPD) and Luria-Bertani (LB) solid media supplemented with increasing concentrations of CuSO_4_. As seen in Figure 1, *P. pastoris* yeast strains are viable up to a 10 mM copper concentration. However, *S. cerevisiae* EBY100, BJ5465, and GRF18 were viable up to 10 mM copper, while *S. cerevisiae* INVSc1 was viable only up to a 5 mM copper concentration. Plates containing only specific solid media without copper were used as the negative control for the viability of strains. *E. coli* strains are more sensitive to high concentrations of copper than yeast. All bacteria strains used are viable up to a 2.5 mM concentration of copper with higher concentrations being lethal for them (Figure 1).

Copper tolerance testing of three *S. cerevisiae* strains (BL7, EL1, and GL7) showed that they can tolerate YPD medium containing copper sulfate only up to a concentration of 6.2 mM [28], compared to the *S. cerevisiae* strains tested in this study (BJ5465, EBY100, and GRF18), which tolerate up to 10 mM copper. The growth rate of bacteria and yeasts was determined by growing strains in specific liquid media containing copper. The copper concentration chosen was 1 mM, at which concentration all strains were viable on plates. The results obtained for the growth rate of bacteria and yeasts by comparing the optical density after 48 h of incubation at 30 °C and 35 °C in copper and copper-free media are shown in Figure 2, Figure 3 and Figure 4.

The growth rate values show that the bacterial strains are around 40% inhibited on media containing 1 mM copper sulfate after 48 h of incubation at 35 °C. These results were obtained against reference cultures grown on media without copper that were considered 100%. Growth inhibition of the *E. coli* OverExpress strain expressing two types of recombinant peroxidase was between 40 and 60 percent.

Interestingly, there is almost no difference between the growth rates of yeast cultures grown on copper and cooper free media (Figure 4). This result indicates that the yeast strains used are more resistant to copper-containing media than the bacterial strains. In the previous experiment, it was also confirmed that yeast cells are viable up to a concentration of 10 mM copper, while bacterial cells are viable only up to a concentration of 2.5 mM. Therefore, the growth rate in liquid media containing copper is in agreement with the results obtained for the copper tolerance test. The mechanisms of copper resistance in bacteria and yeasts involve the biological activity of various copper proteins that are encoded by different genes in their genomes (cytochrome c oxidase, Cu-Zn superoxide dismutase, azurin, laccase, tyrosinase, copper-binding proteins in the inner membrane, and copper chaperones). Therefore, in order to prevent toxicity, copper ions are bound to proteins or stored for subsequent cellular metabolism [29,30]. In bacterial cells, most copper-dependent enzymes are located in the periplasmic space, in membranes, or in the outer space, compared to yeast systems, where copper-dependent enzymes are located intracellularly [30,31]. This characteristic makes the bacteria less resistant to copper-containing media than yeast. *E. coli* uses copper enzymes to maintain its own homeostasis in media containing copper up to 10 µM [32]. *E. coli* strains that tolerate concentrations higher than 10 µM are resistant strains that can be used in the bioremediation of polluted water with copper.

According to scientific literature, yeast strains possess surface adsorption and Cu^2+^ ion biosorption characteristics that are very important in biotechnological processes [28]. However, the bioaccumulation process depends on the yeast strains used. For instance, a high level of 0.1 mM Cu^2+^ inhibited *S. cerevisiae* X2180-1A yeast growth and activity in must [33]. Results of another study showed that *S. cerevisiae* strain BH8 has good tolerance and adsorption of copper ions, and reduces Cu^2+^ concentrations during fermentation in a simple model system mainly through surface adsorption [34].

### 2.2. Copper Accumulation Efficiency

Inductively coupled plasma optical emission spectrometry (ICP-OES) analysis provides more information about the ability of cells to accumulate copper in each strain, including the *E. coli* strains that produced peroxidase enzymes. Cells from 3 mL of cultures, harvested by centrifugation, and washed twice with distilled water were subjected to cell disaggregation in 500 µL 2% HNO_3_ and homogenized. A 300 µL volume of supernatant was subjected to ICP-OES analysis. Nine *E. coli* strains were tested for copper bioaccumulation characteristics.

*E. coli* RIL strain has the best copper accumulation efficiency (1250 times more efficient than the control strain), while the DHα and STAR strains are over 450 times more efficient in retaining copper ions (Figure 5). According to the scientific literature and Stratagene Company, *E. coli* BL21 (DE3) RIL cells contain additional copies of the argU, ileY, and leuW tRNA genes, which recognize the AGA/AGG, AUA, and CUA codons, respectively. This strain possesses protease deficiency and resistance to toxic protein production [35]. There seems to be a relationship between the characteristics of cells that can produce toxic proteins and the efficiency of copper ion accumulation.

Cultures of *E. coli* BL21 (DE3) OverExpress containing different recombinant plasmids carrying peroxidase genes were expressed for 7 h under isopropyl β-D-1-thiogalactopyranoside (IPTG) inducer. Plasmids containing the pelB signal sequence are designed for periplasmic expression, while vectors without this sequence are designed for the internal production of an enzyme [36]. Enzymes were periplasmic and internally expressed, and cultures were further exposed by dilution in copper-containing LB-IPTG medium for another 48 h. Only *E. coli* BL21 (DE3) OverExpress cells containing recombinant plasmids for internal peroxidase expression (Tfu_pET22b no pelB and Tfu_pET21d no pelB) were efficient in copper accumulation (Figure 6). These results suggest that when peroxidase is expressed internally, adaptation, and tolerance mechanisms are activated in the presence of copper and there is an efficient accumulation of the metal [37]. The bacterial strain is used for the production of toxic membrane proteins under the control of the T7 promoter and is also deficient in protease [35].

*S. cerevisiae* strain BJ5465 shows the highest copper accumulation efficiency, followed by *S. cerevisiae* strain INVSc1 (Figure 7). The *S. cerevisiae* yeast appears to have a higher copper accumulation capacity than *P. pastoris*. *S. cerevisiae* BJ5465 is a proteinase-deficient strain used for the production of recombinant proteins [38]. Being a common characteristic of all strains that efficiently accumulate copper, protease deficiency suggests a correlation with copper resistance.

## 3. Materials and Methods

### 3.1. Reagents

All reagents used, such as copper sulfate, yeast extract, tryptone, peptone, sodium chloride, glucose, agar, isopropyl β-D-1-thiogalactopyranoside (IPTG), and ampicillin, were purchased from Carl Roth, Karlsruhe, Germany, or Merck, Darmstadt, Germany.

### 3.2. Recombinant Plasmids

Five recombinant plasmids containing genes for the expression of peroxidase from *Thermobifida fusca*, *Tfu*_pET22b(+) with and without pelB leader sequence, *Tfu*_pET21d(+), and a dye decolorizing peroxidase from *Rhodococcus jostii*, *Dyp*B_pET22b(+) with and without pelB leader sequence have been kindly received from Dr. Raluca Ostafe from the Purdue Institute of Inflammation, Immunology, and Infectious Disease, Molecular Evolution, Protein Engineering, and Production, Purdue University, USA. The blank control plasmid, pET22b(+), was purchased from GenScript, Piscataway, NJ, USA.

### 3.3. Organisms Strains

*Escherichia coli* strains XL10 Gold, DH5α, BL21 (DE3), BL21 (DE3) OverExpress, BL21 (DE3) RP, BL21 (DE3) RIL, BL21 (DE3) Star, BL21 (DE3) Rosseta Gami, and BL21 (DE3) Shuffle were bought from ATCC, USA. *Saccharomyces cerevisiae* EBY100, INVSc1, BJ5465, GRF18, and *Pichia pastoris* X-33, KM71H strains were purchased from Invitrogen, Waltham, MA, USA, or Agilent Technologies, Santa Clara, CA, USA.

### 3.4. Culture Media

Luria–Bertani (LB) medium containing 0.5% (*w*/*v*) yeast extract, 0.5% (*w*/*v*) NaCl, and 1% (*w*/*v*) tryptone at pH 7.40 was used to grow the bacterial strains. Yeast-Peptone-Dextrose (YPD) medium containing 1% (*w*/*v*) yeast extract, 2% (*w*/*v*) peptone, and 2% (*v*/*v*) glucose was used for yeast cell culture. Both media were prepared with or without 2% (*w*/*v*) agar and autoclaved at 121 °C for 30 min. Stock solutions of 2 mM copper sulfate were prepared in LB or YPD media. In addition, LB medium containing ampicillin (0.2 mg/mL) and 2 mM IPTG was prepared. The solutions were then sterilized by filtration through hydrophilic polyvinylidene fluoride (PVDF) membranes with 0.22 µm pores. 

### 3.5. Copper Tolerance Assay of Yeast and Bacterial Strains

Different strains of *E. coli*, *S. cerevisiae,* and *P. pastoris* (see Section 3.3) were grown on LB or YPD solid media containing different concentrations of copper sulfate: 0 (negative control), 0.5, 1.0, 2.5, 5.0, 10.0, 25.0, and 50.0 mM. Each plate was inoculated with 5 µL of each preculture. Plates were incubated for two days at 35 °C or 30 °C. Copper tolerance was given by experiments related to the viability of bacteria and yeasts on plates. 

### 3.6. Transformation of E. coli with Recombinant Peroxidase Plasmids

Recombinant plasmids containing peroxidase genes were transformed into the *E. coli* BL21 (DE3) OverExpress strain according to the protocol described by Miller et al. [39].

### 3.7. Growth Conditions of Microorganisms

Two simultaneous experiments were conducted on the growth rate of microorganisms. First, *E. coli*, *S. cerevisiae*, and *P. pastoris* strains were cultured in liquid media. The growth of microorganisms was started at an optical density of 600 nm (OD 600 nm) of around 0.1, then the cultures were incubated for 24 h at 35 °C or 30 °C. Second, 200 µL of five recombinant *E. coli* strains precultures containing plasmids with peroxidase genes (see Section 3.2) were first inoculated in 10 mL LB containing 100 µg/mL ampicillin and then incubated for 2 h at 35 °C at 300 rpm. In the exponential phase (OD 600 nm around 1.0), the expression of recombinant peroxidases was induced by adding to the cultures IPTG 1 mM. These were further incubated at 35 °C, 300 rpm for 7 h.

To each 3 mL from each culture, 3 mL of specific medium, LB, YPD, and LB-ampicillin-IPTG containing 2 mM copper sulfate were added. All cultures were incubated for 48 h, 300 rpm, 35 °C (bacteria), and 30 °C (yeast). The OD 600 was measured after 48 h of incubation, as well. The series of cultures that have not contained copper were used as negative control samples.

### 3.8. Cell Sample Preparation

After 48 h of growth, 3 mL of each culture was centrifuged for 2 min at 11,000× *g*. The collected cells were resuspended twice in 1.5 mL of distilled water and centrifuged under the same conditions. After removal of the supernatant, cells were resuspended in 500 µL of 2% HNO_3_ solution and left at room temperature for 72 h. The cell debris was additionally subjected to homogenization for 30 s at 20,000 rpm using a Miccra D-1 homogenizer from Germany. Samples were centrifuged for 10 min at 11,000× *g*. The supernatant from each cell sample was subjected to analysis by inductively coupled plasma optical emission spectrometry (ICP-OES).

### 3.9. ICP-OES Analysis

A volume of 300 µL of each cell sample was analyzed using an inductively coupled plasma optical emission spectrometer (ICP-OES), model Spectro Arcos, Germany. The plasma was stabilized and optically profiled according to the manufacturer’s recommendations (ISO 11885:2007—Water Quality). The accumulated copper in the cells was determined at a wavelength of 324.754 nm. Operating conditions were as follows: forward power of 1450 W, coolant flow of 13.0 L/min, nebulization flow of 0.75 L/min, an auxiliary flow of 1.0 L/min, and a sample aspiration rate of 2 mL/min. All chemicals used were of high purity. Certified reference materials (VHG Labs, QWPTM-15) were used for quality control of chemical analyses.

## 4. Conclusions

The growth of the bacterial strains used in these experiments was inhibited upon exposure to 1 mM copper, while yeast cultures were unaffected at this concentration. These results were consistent with the YPD and LB solid medium rapid assays in terms of copper tolerance. ICP-OES analysis showed that the *E. coli* RIL strain had the best copper accumulation efficiency (1250 times more efficient than the control strain), while the DHα and STAR strains were about 450 times more efficient. *E. coli* BL21 (DE3) OverExpress cells that internally expressed peroxidase had the best copper accumulation efficiency. The *S. cerevisiae* BJ5465 yeast strain is the most efficient in copper accumulation of the six yeast strains used, accumulating over 400 times more copper than the negative control strain. Common genetic characteristics of the strains that showed efficiency in copper accumulation involve protease deficiency and resistance to toxic protein production, which is a possible key to successful heavy metal accumulation. Through acidic exposure and homogenization, all components of the cells were disintegrated. As a result, the method presented in this study effectively demonstrates both the efficiency of copper ion accumulation and the capacity of the strains’ cell walls to bind copper. The bioengineered yeasts and bacteria, as well as various strains expressing metal-dependent enzymes used for the first time, are successful candidates for copper accumulation to remove above-limit copper concentrations from water. A promising objective for the management of heavy metal-contaminated water remediation is developing genetically engineered strains that not only accumulate metals but also express proteins capable of binding multiple metal ions.

## Figures and Tables

**Figure 1 ijms-24-07575-f001:**
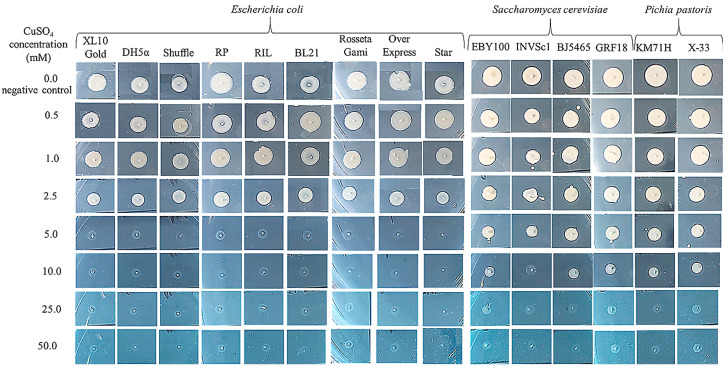
Viability testing of various strains of bacteria and yeast on media with different concentrations of copper sulfate (0 mM—negative control, 0.5, 1.0, 2.5, 5.0, 10.0, 25.0, and 50.0 mM).

**Figure 2 ijms-24-07575-f002:**
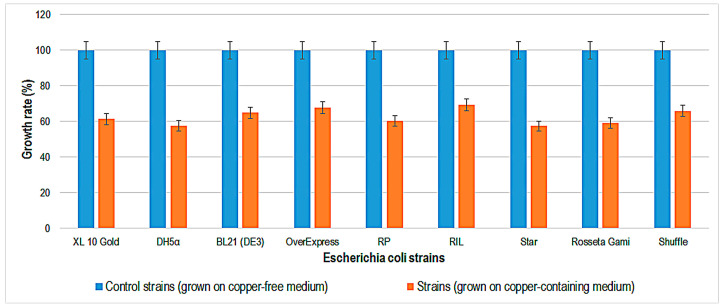
Growth rate (%) for different *Escherichia coli* strains, after 48 h of incubation, at 35 °C, in Luria–Bertani medium without and with copper 1 mM.

**Figure 3 ijms-24-07575-f003:**
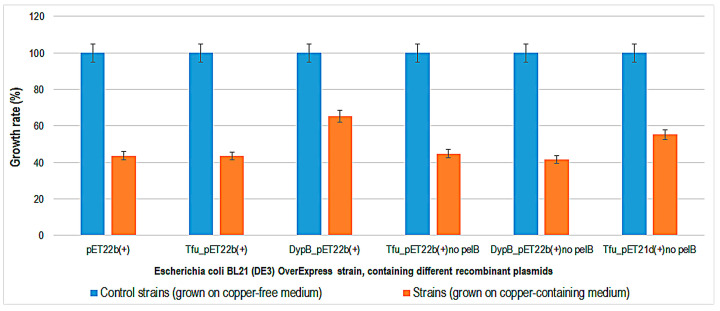
Growth rate (%) for *Escherichia coli* OverExpress containing different recombinant plasmids, after 48 h of expression in Luria-Bertani-isopropyl β-D-1-thiogalactopyranoside-ampicillin medium without and with copper (1 mM).

**Figure 4 ijms-24-07575-f004:**
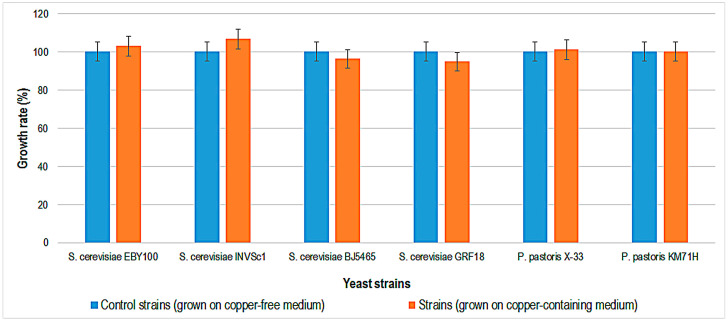
Growth rate (%) for different yeast strains, after 48 h of incubation, at 30 °C, in Yeast-Peptone-Dextrose medium without or with copper (1 mM).

**Figure 5 ijms-24-07575-f005:**
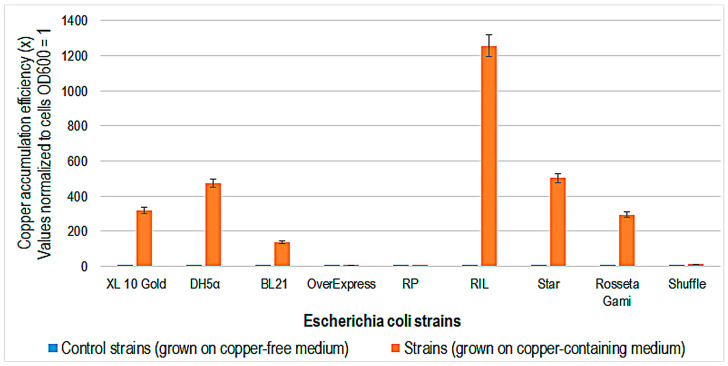
Copper accumulation efficiency (x) in different strains of *Escherichia coli* after 48 h growth in copper, containing Luria–Bertani medium at 35 °C. Control strains were grown on copper-free medium.

**Figure 6 ijms-24-07575-f006:**
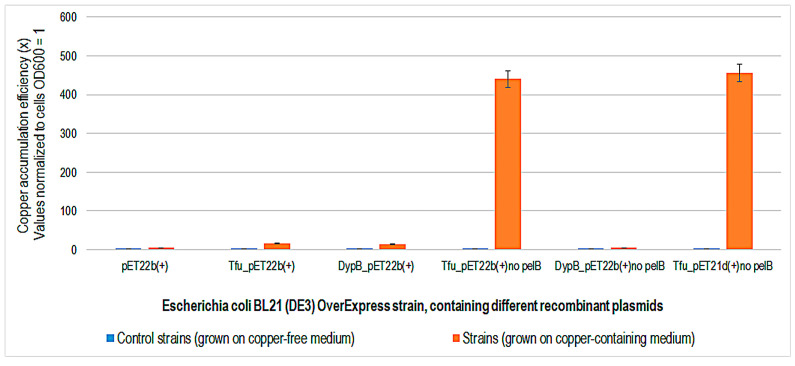
Copper accumulation efficiency (x) in *Escherichia coli* BL21 (DE3) OverExpress containing different recombinant plasmids expressing peroxidase after 48 h of expression in copper-containing Luria-Bertani-isopropyl β-D-1-thiogalactopyranoside -ampicillin medium at 35 °C. Control strains were grown in copper-free medium.

**Figure 7 ijms-24-07575-f007:**
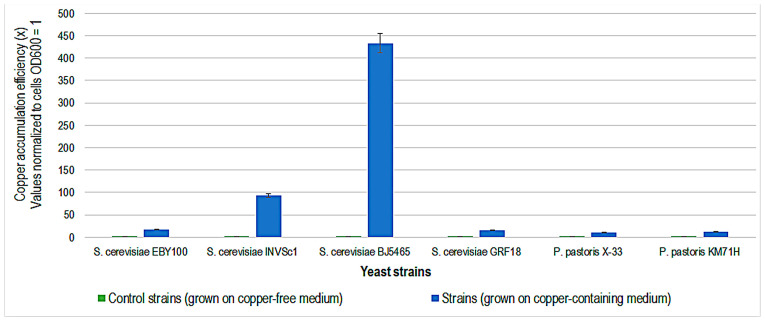
Copper accumulation efficiency (x) of different yeast strains after 48 h of incubation at 30 °C, in copper-containing Yeast-Peptone-Dextrose medium. Control strains were grown in a copper-free medium.

## Data Availability

Not applicable.

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
