# Peer review of "Copper Accumulation Efficiency in Different Recombinant Microorganism Strains Available for Bioremediation of Heavy Metal-Polluted Waters"

_ijms, 2023, doi:10.3390/ijms24087575_

Round 1
Reviewer 1 Report
This manuscript is interesting for green chemistry community. The topic is definitely original and actual. The aim of this research was to investigate the bioremediation conditions of copper in synthetic water. In this study, copper ions accumulation efficiency was determined using various genetically modified strains of Saccharomyces cerevisiae (EBY100, INVSc1, BJ5465, GRF18), Pichia pastoris (X-33, KM71H), Escherichia coli (XL10 Gold, DH5α, and six types of BL21 (DE3)), and Escherichia coli BL21 (DE3) OverExpress expressing two different peroxidases. Viability tests of yeast and bacterial strains showed that bacteria are viable at copper concentrations up to 2.5 mM and yeasts up to 10 mM. Optical emission spectrometry with inductively coupled plasma analysis showed that the tolerance of bacterial strains on media containing 1 mM copper was lower than the tolerance of yeast strains at the same copper concentration. The E. coli BL21 RIL strain had the best copper accumulation efficiency, 1250 times more efficient than the control strain. Yeast strain S. cerevisiae BJ5465 was the most efficient in copper accumulation out of a total of six yeast strains used, accumulating over 400 times more than the negative control strain. In addition, E. coli cells that internally expressed recombinant peroxidase from Thermobifida fusca were able to accumulate 400-fold more copper than cells that produced periplasmic recombinant peroxidases. The aim of the study is clear and the authors provided adequate information on how they conclude their results. The references are relevant and generally recent and include appropriate studies. It is clear what is already known about the topic. The research question is clearly outlined and justified. The process is valid and the variables are defined appropriately. The introduction provides sufficient background. The research methodology is adequate and modern. The results are clearly presented. The conclusions supported by the data. The manuscript good illustrated and interesting to read. English language and style are fine. I have only couple of minor suggestions:
- There are some problems with references (pages 3, 4, 6) - please, fix it.
- It would be a good idea to cite in introduction some relevant papers about other promising adsorbent materials based on coordination compounds: Inorganic Chemistry 2020 59 (23), 17320-17325; Polyhedron 2018 139, 282-288; Journal of catalysis 2020 385, 324-337.
- Some more detailed perspectives regarding the future research could be formulated in conclusions section.
Overall, this nice manuscript could be accepted for publication after minor revisions.
Author Response
Dear Reviewer, thank you very much for your kind and helpful comments. Each of your requests was addressed as explained below. Your comments were written in black, and our responses were in blue. Minor English revisions were performed throughout the entire manuscript. The changes that have been made to the original manuscript (ijms-2339059) can be observed using the track changes function.
Please see the attachment!

Reviewer 2 Report
The authors present results of cytotoxicity and bioaccumulation studies conducted on yeast and E. coli engineered to overexpress peroxidase enzymes conferring tolerance to high concentrations of copper. The strains are intended for use in bioremediation of water containing high levels of Cu. They report that the engineered strain so tolerance to mM levels of Cu in media and that two of the recombinant strains have increased abilities to accumulate Cu versus control organisms. The article is generally well-written and scientifically sound. This reviewer has the following comments:
1) In the Results and Discussion section, there is a recurrent "Error! reference not found" notation. Please address.
Author Response

(The authors gave the same response as above.)

Reviewer 3 Report
The focus of the manuscript is on bioremediation of Cu with some fungi and bacteria species.
The increasing pollution of heavy metals in water has become a serious global issue. The bioremediation of heavy metal-polluted water through the use of genetically modified microorganisms has been shown to be a promising solution. This research investigated the efficiency of copper accumulation by various recombinant microorganism strains, including Saccharomyces cerevisiae, Pichia pastoris, and Escherichia coli. The results demonstrate the ability of these strains to tolerate and accumulate copper ions, with E. coli BL21 RIL and S. cerevisiae BJ5465 showing the highest accumulation efficiency.
1. Line 23; "the E. coli BL21 RIL strain had the best copper accumulation efficiency,...." Mention the values.
2. After reading the introduction, I was unable to identify the novelty in your study. Can you clarify how your study differs from previous research and what gap in knowledge it addresses?
3. Your manuscript contains an error. You assumed that all removal of Cu was due to accumulation, but there is a portion of Cu that would be removed through biosorption. How did you monitor this? Typically, dead biomass is used for biosorption by contacting it with the metals.
Author Response

(The authors gave the same response as above.)

Reviewer 4 Report
BRIEF SUMMARY
Interesting work, but I believe that some changes must be done before it could be ready for publication in this journal. I present my remarks/corrections below.
SPECIFIC COMMENTS
1. Line 14: It is not necessary to repeat the e-mail of the corresponding author. Just write “*Correspondence” and delete the rest text of lines 14.
2. Line 16: Avoid words like “we”, “our”, “us”, as they sound selfish. Replace “In our study” with “In the present study”.
3. Figures: There must be an explanation/reference of its text inside the text. The Figure must be appeared after this explanation.
4. CONCLUSIONS: This paragraph must be extended. The authors here should point the original parts of their work and their contribution in the existing literature. Why is this work important? Which are the new results/conclusion?
5. REFERENCES: Please check that all the references (names of authors and journals, titles, numbers of volumes and pages) are correct. Also check if you have followed the instructions of the journal about how to write the references and ensure that all the references are part of this work. Is it necessary to write the DOI for each one of them? I also find the number of 49 references for a work of totally 11 pages too big, did you really use all of them?
Author Response

(The authors gave the same response as above.)

Round 2
Reviewer 3 Report
Reviewers' comments have been addressed.
Reviewer 4 Report
Accept in present form